# A Cross-Sectional Study Using STROBE Platform to Examine Sleep Characteristics, Mental Health and Academic Performance of Female Applied Medical Sciences Students in the Southwest of Saudi Arabia

**DOI:** 10.3390/bs13060451

**Published:** 2023-05-30

**Authors:** Vandana Esht, Mohammed M. Alshehri, Marissa J. Bautista, Abhishek Sharma, Meshal Alshamrani, Aqeel M. Alenazi, Bader A. Alqahtani, Ahmed S. Alhowimel, Ali Hakamy, Siddig Ibrahim Abdelwahab

**Affiliations:** 1Department of Physical therapy, College of Applied Medical Sciences, Jazan University, Jazan 45142, Saudi Arabia; vandanaesht@jazanu.edu.sa (V.E.); moalshehri@jazanu.edu.sa (M.M.A.); mbautista@jazanu.edu.sa (M.J.B.); 2Department of Physiotherapy, Arogyam Institute of Paramedical and Allied Sciences, Affiliated to H.N.B. Uttarakhand Medical Education University, Roorkee 247661, Uttarakhand, India; 3Department of Pharmaceutics, College of Pharmacy, Jazan University, Jazan 45142, Saudi Arabia; malshamrani@jazanu.edu.sa; 4Health and Rehabilitation Sciences, Prince Sattam Bin Abdulaziz University, Al-Kharj 16278, Saudi Arabia; aqeel.alanazi@psau.edu.sa (A.M.A.); ba.alqahtani@psau.edu.sa (B.A.A.); a.alhowimel@psau.edu.sa (A.S.A.); 5Department of Respiratory Therapy, College of Applied Medical Sciences, Jazan University, Jazan 45142, Saudi Arabia; aohakamy@jazanu.edu.sa; 6Medical Research Centre, Jazan University, Jazan 45142, Saudi Arabia; sadiqa@jazanu.edu.sa

**Keywords:** academic achievement, college students, sleep duration, quality of sleep, well-being

## Abstract

**Background:** Medical students are expected to excel in their academics. Hence exposing them to a certain amount of strain may sometimes cause sleep disruptions. The purpose of this study was to determine the sleep quality among Saudi Applied Medical Sciences students and its potential connections with their academic performance and mental health. **Methods:** This online cross-sectional questionnaire-based study was conducted at Jazan University’s College of Applied Medical Sciences in Saudi Arabia. The Pittsburgh Sleep Quality Index (PSQI), a known indicator of sleep quality, a validated mental health measure; Depression Anxiety Stress Scales-21 (DASS-21), and Academic Performance Scale (APS) with 89 internal consistencies were included in the questionnaire. The cumulative grade point average (GPA) was also used as a covariate to evaluate students’ academic success. **Results:** 112 people responded (response rate = 93%), and 105 of them presented comprehensive information about their backgrounds, way of life, academic standing, sleep patterns, and mental health. Participants’ average GPA and APS scores were 4.23 ± 0.52 and 33.16 ± 5.63, respectively. The mean global PSQI score was 6.47 with 2.34 of standard deviation (SD). The majority of individuals (60%) had poor sleep quality particularly due to abnormal sleep latency and lesser sleep duration, as determined by their PSQI score. The prevalence rates for depression, anxiety, and stress were higher; 53%, 54% and 40%, respectively. Both depression and anxiety were substantially correlated with poor sleep quality (*p*-value = 0.008, *p*-value = 0.01, respectively). Sleep quality had no significant effect on GPA while global PSQI and depression were significantly negatively correlated with an APS score of participants (*p*-value = 0.007 and 0.015, respectively). **Conclusions:** Higher rates of poor sleep quality and psychologically negative emotions were prevalent. Unhealthy sleep patterns were linked to increased levels of anxiety and depression. Self-perceived academic performance was negatively impacted by inadequate sleep and negative emotions, despite the fact that the GPA results were unaffected.

## 1. Introduction

The psychological well-being of students is a significant area of investigation, receiving more and more attention globally [1]. Positive mental health protects against health risk behaviors, with low mental health leading to dangerous behaviors [2]. A significant amount of evidence also demonstrates that people who are depressed and hopeless tend to be less physically active [3] and tend to experience more negative emotions, such as suicidal thoughts [4]. Suboptimal sleep has been negatively linked with the psychological well-being of an individual as sleep plays a crucial role in regulating emotions, mood, and overall mental well-being. When individuals do not receive enough quality sleep or experience disruptions in their sleep patterns, it can have detrimental effects on their psychological health [5]. A large sample of university students from 16 countries reported that excessive (>9 h) or insufficient (<7 h) sleep was linked to an increased risk of social isolation, implying an inverted U-shaped relationship between psychological well-being and sleep duration [6].

In the last decade, a number of studies on the interactions between subjective sleep and the academic achievements of university students have emerged, with a positive relationship between sleep quality and academic scores being reported [7,8,9]. Poor sleep quality and insufficient sleep were found to be significantly associated with poorer academic performances [10]. In addition, a substantially higher proportion of students with disturbed sleep and average exam marks was reported as compared to students with good marks [11]. Departing from these studies, there is no relationship between sleep quality and academic success [12]. The time it takes to fall asleep and wake up affects academic performance more than the duration of sleep does [13]. It was also observed that high academic performance is associated with shorter sleep among young adults between 20 and 21 years of age, including university students [14]. In short, recent findings reported in the literature on the relationship between sleep and academic performance have not fully arrived at a consensus [15].

Factors such as difficulty falling asleep and staying asleep are commonly associated with poor sleep quality [16,17]. Research has demonstrated that poor sleep quality has significant implications for mental health, including an increased risk of conditions, such as depression and anxiety [18,19]. This issue has gained significant attention in public health, considering its impact on overall well-being. Young adults, including university students, are particularly vulnerable to experiencing poor sleep quality. This may be attributed to various factors, such as disrupted biological sleep rhythms due to stress related to future prospects and employment, late-night computer work, academic workload, social integration demands, and environmental noise [20,21,22,23]. These factors can contribute to irregular sleep patterns and disturbances in sleep quality among university students.

Studying sleep characteristics among young university students in Saudi Arabia to determine their sleep quality is an important research endeavor. Additionally, investigating the association between sleep and academic achievements in the southwestern regions of Saudi Arabia, where there is a scarcity of research on this topic, can provide valuable insights into the changing sleep patterns of young individuals [24]. A trend analysis of sleep as it relates to student achievement is needed. Further, there are important methodological challenges in the current literature, such as a lack of representative samples [25]. Having a representative sample gives outcomes that are more likely to resemble the population in general and yield more trustworthy and relevant results. Inferences made from representative samples are, furthermore, generalizable and useful for educational policy.

To address the current research gaps, we utilize the most recent data available from university students. The present study had two main objectives. The first was to observe sleep characteristics and to understand how psychological well-being and the effective components of subjective well-being are related to the self-reported sleep of full-time undergraduate students in the College of Applied Medical Sciences at Jazan University, Saudi Arabia. The targeted endpoint was a model for subjective sleep quality that would reveal the nature of these relationships (e.g., linear versus quadratic) and how the different dependent variables might interact. Nevertheless, due to the contrary findings of the effects of sleep on academic performance in the previous literature, the second primary objective of our study was to find out the association between sleep quality and academic performance of university students. 

## 2. Methods

### 2.1. Study Design and Participants

A cross-sectional correlation study was conducted to evaluate the association among sleep quality, mental health, and academic performance among the female students of the Applied Medical Sciences College at the Jazan University who are aged 18–25 within the Jazan region of Saudi Arabia. Data was collected from females in the Applied Medical Sciences College. Hence, only female participants were included in this study. This study was accepted and approved by the standing committee for scientific research at the Jazan University (HAPO-10-Z-001), Jazan, Saudi Arabia (reference number: REC-44/07/523). STROBE guidelines as shown in Appendix A (Table A1) were followed to conduct the survey.

Using the sample size formula for the survey (cross-sectional) prediction model, the sample size was determined to be 84 students [26]. The parameters used for sample size calculation were the standard deviation of outcome from the Pittsburg Sleep Quality Index (PSQI), SD = 2.8 at 95% confidence interval, and absolute error or precision of d = 0.6 [27]. At a 40% non-response rate assumption, a total sample size of 120 was projected [28]. From the female portion of the College of Applied Medical Sciences at Jazan University, three streams were chosen at random. The selected departments included the Department of Medical Laboratory Technology (MLT), the Clinical Nutrition Department (CN), and the Department of Physical Therapy (PT). The survey was sent to 120 randomly chosen enrolled college students via an online Google form, and a bar code was also generated to access the questionnaire. The survey began with a succinct explanation of the study’s purpose, followed by a thorough-informed consent that covered the subjects’ rights, the perceived risks and rewards of participation, and the steps taken by the researchers to protect the confidentiality of their personal data. Only after they consented to this informed consent were the students able to view the survey. A participant completed the survey in around 10 min. Demographic/characteristics of the participants including age, gender, marital status, ethnicity, race, medical history, area of living, and caffeine consumption were obtained. The cumulative grade point average (GPA) score from the last semester of each participant was recorded. The academic performance scale (APS) by Carson Birchmeier, Emily Grattan, Sarah Hornbacher, and Christopher McGregory of Saginaw Valley State University with an internal consistency of 0.89 was used to monitor how students perceived their own academic performance. Sleep quality using the Pittsburgh Sleep Quality Index (PSQI), sleep quantity, and habitual sleep posture were assessed [29]. The PSQI assesses several aspects of sleep quality, including subjective sleep quality, sleep latency (the time it takes to fall asleep), sleep duration, habitual sleep efficiency (the ratio of time spent asleep to time spent in bed), sleep disturbances (e.g., waking up during the night), use of sleeping medication, and daytime dysfunction (how sleep problems affect daily functioning) [27,30]. By measuring these different components, the PSQI provides a holistic view of sleep quality and can help inform clinical decisions. It offers a standardized and reliable method for assessing sleep quality and identifying potential sleep problems in individuals across different populations [30].

Mental health was assessed using the Depression Anxiety Stress Scales-21 (DASS-21), which is a 21-item self-report inventory measuring the severity of depression, anxiety, and stress in adolescents and adults [31]. It included depression, anxiety, and stress as subscales. Each subscale included seven items, resulting in a total of 21 items. Respondents rate the severity of each symptom over the past week on a 4-point Likert scale ranging from 0 (does not apply at all) to 3 (applies very much or most of the time). Higher scores on each subscale indicate higher levels of depression, anxiety, or stress. The DASS-21 has been extensively used in research and clinical settings to assess the symptomatology associated with depression and anxiety. It provides a convenient and efficient way to measure these constructs simultaneously and has demonstrated good reliability and validity [32]. Students with known untreated sleep disorders (such as sleep apnea or restless leg syndrome), severe chronic pain that highly interferes with sleep, such as severe low back pain, pregnancy, speech deficits, significant uncorrected visual, or auditory impairment), and neurological diseases (e.g., traumatic brain injury, and stroke), bipolar affective disorder, seizure disorder, cardiovascular illness, systolic blood pressure >140 mmHg, or diastolic blood pressure <60 mmHg, respiratory disease, and participants unable to fully follow the instructions were excluded from this study. A total of 112 students (response rate = 93%) answered the survey, however, five of them did not match the requirements (one student was >25 years old and another four reported chronic medical illnesses). Two participants were eliminated due to missing data. As a result, 105 students were included in the analyses with complete data for all variables.

### 2.2. Statistical Analysis

The SPSS software version 21 for Windows was used to conduct the statistical analysis. The 5% level of significance was set. For continuous variables, means and standard deviations (SD) were used to summarize the sample’s characteristics, while for categorical variables, frequencies and percentages were used. For continuous variables, an independent sample *t*-test was utilized, whereas categorical variables were subjected to a Chi-square test. The Pearson’s correlation coefficient was used to assess the linear relationship between selective variables.

## 3. Results

### 3.1. Socio-Demographic Characteristics

Female students constituted 100% of the sample. The sample’s mean age was 20.8 years (standard deviation; SD = 1.35, range = 18–25 years), mean weight was 54.69 kgs (SD = 13.17), mean height was 156.96 cm (SD = 6.11) and mean BMI was 22.26 (SD = 4.5). In total, 84% of individuals were unmarried at the time. CN, PT and MLT Departments constituted 34%, 36%, and 30% of the respondent pool, respectively. The 1–2, and 3–4 years students made up 45.7% and 54.2% of the sample, respectively. In total, 3% of the pupils said they smoked. In addition, 11% of participants lived with their families, 39% shared a room with a roommate, and 48% lived alone in their room. The associations between sleep quality according to the PSQI scores and demographic variables (age, weight, height, BMI, and marital status) of participants showed no significant differences among good and poor sleepers (*p* < 0.05). Years 1 and 2 students were significantly higher in the poor sleeper category as compared to good sleepers (*p*-value = 0.009). A summary comparing the demographic and academic characteristics among good and poor sleepers is represented in Table 1.

### 3.2. Academic Performance

The academic performance scale mean scoring of participants was 33.16 ± 5.63, with a range = 9–40. According to the scale interpretations, 61% of respondents had excellent performance. Ninety percent of students reported attending classes on a regular basis, and seventy-nine percent said they paid attention in class and were always prepared for group discussions. In total, 68% of the students say they enjoyed doing their homework and were more assertive when faced with challenging tasks, and 82% of the students wished to score well in all of their subjects. The average GPA of the sample was 4.23 (SD = 0.52), and 35% of participants had either an excellent or exceptional grade descriptions (Table 2).

### 3.3. Subjective Sleep Quality (PSQI), Sleep Quantity and Sleeping Position

The global PSQI score ranged from 1 to 13 (mean = 6.47, SD = 2.8). According to global PSQI scoring, overall sleep quality was good among 40% of the participants (global PSQI scoring of ≤5). According to the Pittsburgh Sleep Quality (PQSI) Index, subjective sleep quality was good among 83.8% of the participants, 11.4% had medium to poor sleep, while 4.8% of the respondents had extremely poor sleep. Sleep latency was normal among 41.90% of the respondents between 10 and 20 min (Jung DW et al., 2013), and 30.5% of the participants had night sleep duration within the recommended levels (7 to 9 h per night) (Hirshkowitz M et al., 2015). In total, 6% of the participants had a naptime of > 2 h in duration during the day, 47.6% had normal sleep efficiency, and 57% of the participants preferred a side-lying position over a supine or prone lying position for sleeping. In addition, 5% of participants had taken sleep medications in the past month. Only 2% of the total participants reported sleep disturbances three or more times a week and 5% had daytime dysfunctions three or more times a week due to poor sleep.

### 3.4. Depression Anxiety Stress Scales (DASS-21)

The DASS-21 psychological distress score and PQSI sleep quality score had a statistically significant linear connection (*p*-value = 0.0001). The participants’ depression scores ranged from 0 to 20 (mean = 6.07, SD = 5.05), with 31.4% of the participants having mild-to-moderate depression and 21.82% having severe-to-extremely severe depression. The anxiety scores ranged from 0 to 18 (mean = 5.75, SD = 5.15), with 25.6% having mild-to-moderate anxiety and 31.41% having severe-to-extremely severe anxiety. Lastly, the stress scores ranged between 0 and 18 (mean = 6.70, SD = 5.19), with 25.7% having mild-to-moderate stress and 17.2% having severe-to-extremely severe stress (Table 3).

### 3.5. Sleep Quality

The sleep quality, according to the PSQI questionnaire, was assessed based on two options. A total of 42 participants (40%) reported having good sleep quality, defined as a score of five or less. On the other hand, 63 participants (60%) indicated poor sleep quality, with a score higher than five. Participants’ subjective sleep quality was evaluated. Among the respondents, 41 individuals (39%) reported very good sleep quality, a slightly higher number than the 47 participants (44.8%) who expressed fairly good sleep quality. Additionally, 12 participants (11.4%) reported fairly bad sleep quality, while 5 participants (4.8%) indicated very bad sleep quality.

### 3.6. Sleep Latency and Duration

Sleep latency refers to the time it takes to fall asleep, to which 13 individuals (12.4%) reported a sleep latency of less than 15 min. The majority, 51 participants (48.6%), had a sleep latency between 16 and 30 min. Furthermore, 33 participants (31.4%) indicated a sleep latency between 31 and 60 min, while 8 participants (7.6%) reported a sleep latency exceeding 60 min. The duration of sleep was categorized into four options. Among the respondents, 32 individuals (30.5%) reported sleeping for 7 h. Additionally, 40 participants (38.1%) indicated a sleep duration of 6–7 h. Furthermore, 29 participants (27.6%) reported sleeping for 5–6 h, while 4 participants (3.8%) reported sleeping less than 5 h.

### 3.7. Sleep Efficiency and Disturbances

Sleep efficiency, representing the percentage of time spent asleep out of the total time spent in bed, was classified into four categories. Among the participants, 52 individuals (49.5%) reported a sleep efficiency greater than 85%. Moreover, 33 participants (31.4%) had a sleep efficiency between 75% and 84%. Additionally, 15 participants (14.2%) reported a sleep efficiency between 65% and 74%. Lastly, five participants (4.8%) indicated a sleep efficiency lower than 65%. The frequency of sleep disturbances was assessed through four options. Among the participants, 11 individuals (10.5%) reported not experiencing sleep disturbances during the past month. Furthermore, 62 participants (59.0%) reported experiencing sleep disturbances less than once a week. Moreover, 30 participants (28.6%) indicated experiencing sleep disturbances once or twice a week. Lastly, two participants (1.9%) reported experiencing sleep disturbances three or more times a week.

### 3.8. Use of Sleep Medications

The usage of sleep medication was evaluated, for which 90 individuals (85.7%) reported not using sleep medication during the past month. Additionally, five participants (4.8%) indicated using sleep medication less than once a week. Furthermore, four participants (3.8%) reported using sleep medication once or twice a week, while six participants (5.7%) reported using sleep medication three or more times a week. The occurrence of daytime dysfunction was assessed through four options. Among the participants, 31 individuals (29.5%) reported not experiencing daytime dysfunction during the past month. Moreover, 46 participants (43.8%) indicated experiencing daytime dysfunction less than once a week.

## 4. Discussion

To promote better academic performance and mental functioning, it is important for students to prioritize healthy sleep habits. Establishing a consistent sleep schedule, creating a sleep-friendly environment, and practicing good sleep hygiene (e.g., avoiding electronic devices before bed and limiting caffeine intake) can contribute to improved sleep quality. In turn, better sleep can enhance cognitive abilities, concentration, memory, and overall academic performance. The amount and quality of sleep a person obtains over the course of a 24-h period is crucial for their physical, as well as emotional, well-being [28]. The present research was carried out to study the sleep characteristics of applied medical science college students in the southwest region of Saudi Arabia, and to assess their mental health and its subsequent correlation with their academic performance. Increasing awareness and enhancing public knowledge of the benefits of regular sleep and thus, in turn, eventually increasing the proportion of individuals who obtain optimal sleep is in line with the national health goals represented in the Saudi Vision 2030 to achieve improved health, wellness, productivity, and quality of life [33].

### 4.1. Findings from this Study

The results of this study found a high prevalence of poor-quality sleep (60% of total respondents) among youngsters according to the global PSQI scores. However, the perceived subjective poor-sleep quality was much lower (16% of total respondents) The demographic variables (age, weight, height, BMI, and marital status) had no significant influence on the quality of sleep (Table 1). Abnormal sleep latency (normal between 10 and 20 min) (Jung DW et al., 2013) and lesser sleep duration (normal 7 to 9 h of night sleep a day) [24] were among the major components resulting in higher PSQI scores, which in turn lead to a compromised quality of sleep. Due to circadian variations and homeostatic sleep systems, pubertal alterations in the second decade of life have an impact on the sleep and wake timings. Consequently, when young adults attempt to match their naturally delayed schedule with the demands of regular society schedules, such as college and business hours, they may experience sleep loss and excessive daytime drowsiness [34]. University students are said to be chronically sleep-deprived [35]. Poor-sleep quality had been linked to negative emotions, such as anger, bewilderment, despair, and stress [36]. A high number of students were associated with negative psychological emotions in the present study (Table 3). Similar findings had been observed, with both inadequate sleep and poor sleep quality linked to depression in college students [37]. Less sleep was linked to emotions that were both more negative and less positive [38]. Intriguingly, a U-shaped relationship was reported between happiness and sleep duration in adults, with a prevalence of unhappy people among the short sleepers (≤6 h) and excessively long (≥9 h) sleepers [39]. The prevalence of short sleepers was also high within the present study (69% of total respondents). The findings of the present study also revealed a significant association between depression and anxiety, and negative emotions with poor sleep quality (Table 1).

### 4.2. Sleep Quality and Academic Performance

Unadjusted analyses of academic performance with various sample characteristics showed that sleep quality and academic year were significantly associated. Among students with different academic years, there was a statistically significant variation in the sleep quality scores with year 1 and year 2 students having higher prevalence of poor sleep (*p*-value 0.001) (Table 1). According to the findings, the PSQI scores were statistically, significantly, and negatively correlated with the APS scores *p*-value (Table 2). Higher PSQI scores indicated more deterioration in sleep quality, while APS, a self-perceived academic performance, higher scores indicated a higher perception of performing well academically. Hence, individuals having a poorer quality of sleep perceived a lack of performing well academically. However, no associations of PSQI scores were observed with the GPA scores of participants. Even though the link between sleep and academic achievement has been discussed in the medical literature for a while, the question remains unresolved. Although all three variables were related to academic success (a positive relationship between sleep quality and duration, and a negative association with sleepiness), the effect of sleep quality, sleep duration, and sleepiness on adolescents’ academic performance was very minimal. Using a bivariate analysis of covariance, to examine the relationship between sleep quality, mental health, and academic success, sleep quality and depression were significantly associated with reported APS scores (*p* < 0.05) (Table 2). Academic performance has also been found to be negatively correlated with sleep latency [38]. A high prevalence of abnormal sleep latency had been found in the present study. Lack of adequate sleep also interferes with the function of the brain structures critical to cognitive processes. The most notably impacted structure is the prefrontal cortex, which executes higher brain functions, including language, working memory, logical reasoning, and creativity. The research findings confirm the existing evidence regarding the prevalence of poor sleep quality among university students [40] and its associations with psychological well-being [36]. It is well-established that normal sleep quality is crucial for overall physiological functioning, and when sleep quality deteriorates, it can lead to significant psychological problems [41]. Specifically, poor sleep quality has been linked to an increased risk of depression and anxiety.

These results emphasize the significance of addressing and improving sleep quality among university students, as it can have a positive impact on their psychological well-being. By recognizing the relationship between sleep quality and mental health, interventions and strategies can be developed to promote better sleep habits and ultimately contribute to improved mental health outcomes. The findings of the current investigation demonstrated a direct relationship between depression and anxiety, and sleep quality. According to a research study, major depressive disorder, chronic stress, and anxiety disorders are connected. Severe problems, including insomnia, immune system deterioration, high blood pressure, anxiety, and muscle discomfort, can develop if persistent stress is not managed. The study’s findings, which assessed the levels of stress, anxiety, and depression among students, also revealed a weak but significant link between depression and the participants’ rates of stress and anxiety.

### 4.3. Strength and Limitations

Few studies have explored sleep habits in populations of students who are pursuing health care degrees. The present research provides tangible evidence regarding sleep characteristics and the subsequent association with academic performance and mental health of female applied medical sciences students. This study was limited to several key factors including objective measurements of sleep, sleep consistency, chronotype preference, male gender, and subjective well-being of participants. A future direction pertaining to all the missed factors and development of separate models of subjective and objective sleep quality will lead to a more in-depth investigation.

## 5. Conclusions

It has been concluded from the present study that normal sleep quality is linked to lower levels of mental problems and better academic performance; poor sleep quality is associated with higher levels of negative mental health and poor academic performance. Moreover, the impact appears to be more significant for individuals experiencing poor sleep quality, suggesting that addressing and avoiding poor sleep quality is crucial to reduce the risk of mental problems. Increasing awareness about the importance of good sleep quality among undergraduate students can be beneficial as it can help students recognize the potential impact of poor sleep on their mental well-being and overall health. This awareness can motivate them to prioritize and adopt healthy sleep habits.

Educational initiatives can be developed to provide information and resources on sleep hygiene practices. These may include promoting consistent sleep schedules, creating conducive sleep environments, managing stress effectively, limiting the use of electronic devices before bedtime, and promoting relaxation techniques to enhance sleep quality.

Additionally, providing support systems within educational institutions, such as counseling services or workshops focused on sleep health, can help students address many existing sleep difficulties and develop strategies to improve their sleep quality. Emphasizing the importance of avoiding poor sleep quality and promoting healthy sleep practices can have positive implications for their academic performance, overall well-being, and future success. It is worth noting that specific interventions and strategies may vary depending on cultural and contextual factors, so it is important to tailor these efforts to the unique needs and circumstances of the target population.

## Figures and Tables

**Table 1 behavsci-13-00451-t001:** Comparison of demographic, academic and mental health characteristics of participants among good and poor sleepers according to PSQI (n = 105).

	Good Sleepers Global PSQI ≤5 (n = 42)	Poor Sleepers Global PSQI >5 (n = 63)	*p*-Value
Age in years *	20.67 ± 1.43	20.81 ± 1.31	0.424
Weight in kg *	54.26 ± 12.95	54.97 ± 13.41	0.349
Height in cm *	156.17 ± 5.74	157.49 ± 6.33	0.485
BMI in kg/m^2^ *	22.18 ± 4.63	22.15 ± 5.26	0.654
Marital status ** Married Unmarried	6 (14.2) 36 (85.7)	11 (17.4) 52 (82.5)	0.225 0.088
Cumulative Grade Point Average Score (GPA) *	4.26 ± 0.48	4.20 ± 0.543	0.441
Academic performance scale (APS) score *	34.62 ± 5.02	32.19 ± 5.84	0.773
Department PT ** CN ** MLT **	18 (42.8) 10 (23.8) 14(33.3)	20 (31.7) 26 (41.3) 17(27)	0.746 0.008 0.590
Year ** 1 and 2 3 and 4	15 (35.7) 27 (64.3)	33 (52.3) 30 (47.6)	0.009 0.691
Psychological Negative emotion of DASS-21-Depression	3.50 ± 3.62	7.78 ± 5.17	0.008
Psychological Negative emotion of DASS-21-Anxiety	3.81 ± 3.94	7.05 ± 5.46	0.01
Psychological Negative emotion of DASS-21-Stress	4.52 ± 4.34	8.16 ± 5.22	0.264

* Values expressed as (mean ± SD); independent-Samples t-test was used for comparison between values, ** values expressed as frequency (percent); non-parametric chi-square test was used for comparison between values.

**Table 2 behavsci-13-00451-t002:** Descriptive Academic characteristics, responses of Academic Performance Scale (APS) and Pearson correlation between APS score and selective variables.

S. No.	Question	Options	Frequency and Percentage N (%)
1.	I made myself ready in all my subject	A. Strongly agree B. Agree C. Neutral D. Disagree E. Strongly disagree	44 (41.9) 39 (37.1) 19 (18.0) 2 (0.01) 1 (0.0)
2.	I pay attention and listen every discussion	A. Strongly agree B. Agree C. Neutral D. Disagree E. Strongly disagree	45 (42.8) 38 (36.1) 18 (17.1) 3 (0.08) 1 (0.0)
3.	I want to get good grades in every subject	A. Strongly agree B. Agree C. Neutral D. Disagree E. Strongly disagree	86 (81.9) 13 (12.3) 3 (0.08) 1 (0.0) 2 (0.01)
4.	I actively participate in every discussion	A. Strongly agree B. Agree C. Neutral D. Disagree E. Strongly disagree	31 (29.5) 36 (34.2) 32 (30.4) 5 (0.04) 1 (0.0)
5.	I start papers and projects as soon as they are assigned	A. Strongly agree B. Agree C. Neutral D. Disagree E. Strongly disagree	42 (40.0) 39 (37.1) 17 (16.1) 5 (0.04) 2 (0.01)
6.	I enjoy homework and activities because they help me improve my skills in every subject.	A. Strongly agree B. Agree C. Neutral D. Disagree E. Strongly disagree	38 (36.1) 33 (31.4) 23 (21.9) 7 (0.06) 4 (0.03)
7.	I exert more effort when I do difficult assignments	A. Strongly agree B. Agree C. Neutral D. Disagree E. Strongly disagree	62 (59.0) 34 (32.3) 6 (0.05) 1 (0.0) 2 (0.01)
8.	Solving problems is a useful hobby for me	A. Strongly agree B. Agree C. Neutral D. Disagree E. Strongly disagree	34 (32.3) 33 (31.4) 25 (23.8) 9 (0.08) 4 (0.03)
9.	Total score of APS	A. Excellent performance B. Good performance C. Moderate performance D. Poor performance	64 (61.0) 33 (31.4) 7 (6.7) 1 (1.0)
10.	Total GPA score	A. Exceptional B. Excellent C. Superior D. Very Good E. Above Average F. Good	12 (11.4) 25 (23.8) 38 (36.2) 21 (20.0) 8 (7.6) 1(0.9)
11.	Pearson correlation coefficients (r) for Total score of APS with selected variables	A. PSQI score B. Depression score C. Anxiety score D. Stress score	r = −0.261, *p* = 0.007 r = −0.236, *p* = 0.015 r = −0.162, *p* = 0.10 r = −0.163, *p* = 0.098

**Table 3 behavsci-13-00451-t003:** Psychological negative emotion and descriptive characteristics of responses DASS-21.

S. No.	Question	Options *	Frequency and Percentage N (%)
1.	Mean SD score of negative emotion	Depression Anxiety Stress	6.07 ± 5.05 5.75 ± 5.15 6.70 ± 5.19
2.	Depression	Normal Mild Moderate Severe Extremely Severe	49 (46.6) 8 (7.62) 25 (23.8) 13 (12.3) 10 (9.52)
3.	Anxiety	Normal Mild Moderate Severe Extremely Severe	45 (42.8) 14 (13.3) 13 (12.3) 6 (5.71) 27 (25.7)
4.	Stress	Normal Mild Moderate Severe Extremely Severe	60 (57.1) 12 (11.4) 15 (14.3) 15 (14.3) 3 (2.86)
5.	I found it hard to wind down	0 1 2 3	32 (30.4) 27 (25.7) 26 (24.7) 10 (0.09)
6.	I was aware of dryness of my mouth	0 1 2 3	47 (44.7) 36 (34.2) 14 (13.3) 8 (0.07)
7.	I couldn’t seem to experience any positive feeling at all	0 1 2 3	44 (41.9) 36 (34.2) 17 (16.1) 8 (0.07)
8.	I experienced breathing difficulty (e.g., excessively rapid breathing, breathlessness in the absence of physical exertion)	0 1 2 3	56 (53.3) 29 (27.6) 12 (11.4) 8 (0.07)
9.	I found it difficult to work up the initiative to do things	0 1 2 3	40 (38.0) 45 (42.8) 16 (15.2) 4 (0.03)
10.	I tended to over-react to situations	0 1 2 3	47 (44.7) 26 (24.7) 25 (23.8) 7 (0.06)
11.	I experienced trembling (e.g., in the hands)	0 1 2 3	48 (45.7) 32 (30.4) 15 (14.2) 10 (0.09)
12.	I felt that I was using a lot of nervous energy	0 1 2 3	34 (32.3) 36 (34.2) 22 (20.9) 13 (12.3)
13.	I was worried about situations in which I might panic and make a fool of myself	0 1 2 3	45 (42.8) 32 (30.4) 13 (12.3) 15 (14.2)
14.	I felt that I had nothing to look forward to	0 1 2 3	51 (48.5) 29 (27.6) 16 (15.2) 9 (0.08)
15.	I found myself getting agitated	0 1 2 3	41 (39.0) 40 (38.0) 16 (15.2) 8 (0.07)
16.	I found it difficult to relax	0 1 2 3	42 (40.0) 35 (33.3) 16 (15.2) 12 (11.4)
17.	I felt down-hearted and blue	0 1 2 3	43 (40.9) 28 (26.6) 23 (21.9) 11 (10.4)
18.	I was intolerant of anything that kept me from getting on with what I was doing	0 1 2 3	47 (44.7) 32 (30.4) 20 (0.19) 6 (0.04)
19.	I felt I was close to panic	0 1 2 3	59 (56.1) 23 (21.9) 15 (14.2) 8 (0.07)
20.	I was unable to become enthusiastic about anything	0 1 2 3	41 (39.0) 39 (37.1) 15 (14.2) 10 (0.09)
21.	I felt I wasn’t worth much as a person	0 1 2 3	58 (55.2) 30 (28.5) 7 (0.06) 10 (0.09)
22.	I felt that I was rather touchy	0 1 2 3	53 (50.4) 31 (29.5) 14 (13.3) 7 (0.06)
23.	I was aware of the action of my heart in the absence of physical exertion (e.g., sense of heart rate increase, heart missing a beat)	0 1 2 3	52 (50.5) 25 (23.8) 9 (0.09) 14 (13.3)
24.	I felt scared without any good reason	0 1 2 3	52 (49.5) 30 (28.5) 8 (0.07) 15 (14.2)
25.	I felt that life was meaningless	0 1 2 3	57 (54.2) 25 (23.8) 8 (0.07) 15 (14.2)

Options *: 0: Did not apply to me at all; 1: Applied to me to some degree, or some of the time; 2: Applied to me to a considerable degree or a good part of time; 3: Applied to me very much or most of the time.

## Data Availability

All relevant data supporting this study’s finding are within the manuscript.

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
