# Peer review of "A Cross-Sectional Study Using STROBE Platform to Examine Sleep Characteristics, Mental Health and Academic Performance of Female Applied Medical Sciences Students in the Southwest of Saudi Arabia"

_behavsci, 2023, doi:10.3390/bs13060451_

Round 1

Reviewer 1 Report

The article "A Cross-Sectional Study Using STROBE Platform to Examine Sleep Characteristics, Mental Health and Academic Performance of Female Applied Medical Sciences Students in the Southwest of Saudi Arabia" is an impressive piece that deserves recognition. The introduction is well documented and easy to understand; the methodology clearly describes each step taken (sample size, sample description, informed consent process, data analysis etc.), while results are written in a way that makes them easy to comprehend. Furthermore, the discussion section provides great insight into this topic as well as relevant citations from recent literature sources. All together this article is original and engaging for readers interested on these topics making it a worthwhile read.

If I had to add something to the article it would surely be a comparison with other faculties of health sciences to explore and analyze the respective differences. However, the article in its present state can be a good starting point for further research on the argument and research topic. 
The tables are well presented.

However, to improve some aspects, I would include more bibliographic references.

Author Response

The authors are really thankful to you for providing really valuable feedback to improve the quality of the manuscript.

According to your feedback, we have added more bibliographic references. You can appreciate the changes highlighted in red color in the revised manuscript.

Kind regards

Reviewer 2 Report

Title

It is not clear or justifiable why the particular focus on "female" students is identified in the title of the manuscript when there's no further, direct references in the abstract, introduction or methodology. 

Introduction: 

Page 1, lines 33-34: "Suboptimal sleep has been negatively linked with psychological well being of an 33 individual[1]. A large sample of university students from 16 countries." This is an inconclusive sentence, similar issues regarding the quality of English in the manuscript are present in multiple other areas, a more thorough language revision should be conducted before resubmitting. 

Page 2, lines 51-54: "There is need to study sleep characteristics among young university students of 51 Saudi Arabia to determine their sleep quality and also while there are studies that ad-52 dress the association between sleep and academic achievements internationally, there is 53 a need for new knowledge due to scanty of research assessing this relationship in 54 southwestern regions of Saudi arabia. This is especially true because there seems.." These 'needs' are not clearly identified in your introduction, there's no logical step that was followed to arrive at this conclusion. 

Overall, your introduction needs a major revision. At present, it does not flow well and there is unsubstantiated claims lacking the necessary data/evidence. There are numerous issues related to grammar and syntax as well as sentences that lead to nowhere.

Methods:

You mention STROBE guidelines were followed, there's no STROBE checklist with your submission.

Page 2, line 83: You mention PSQI with no prior mention of the abbreviation.

Page 3, lines 102-103: More information on your instruments such as subscales, scoring and other relevant information would have been important to include here.

There's no mention or justification for the reason behind selecting female-only participants. It is known that some Saudi universities have female-only colleges but that is not mentioned somewhere. Clarifying this point would have been important. 

Results:

Tables 2-3-4 are very hard to read as they contain a very large amount of information running over multiple pages. You should have considered how easy it is to read these tables. 

Results should have been better summarised as currently, they span a large section of the manuscript with no clarity.

Discussion

Page 9, lines 188-189: "Students’ academic performance is vastly influenced by their mental functioning 188 which in turn is dependent on adequate sleep and vice versa." This is a very strong statement, not fully supported by your review of evidence (introduction) or your results. This should be rephrased. 

Page 9, lines 194-197: this is not an appropriate place for this statement, You could have discussed how this work relays on policy and other initiatives later on in your discussion, not at this stage. 

Page 10, line 222: You mention "predictors" when you only run correlations. Further, more robust analyses should have been in place to be able to allow for such statements. Please revise and rephrase. This entire section is based on an unjustified premise. 

There is no limitations (and strengths) section, that is a major omission.

There is no clear conclusion, that is a major omission. 

Generally, this paper is not at a "publication-ready" stage and needs significant work before it could be resubmitted to this or another journal.  

As indicated in my review, there are multiple instances where English language could be improved. There are sentences that are completely out of place (see introduction) with no logical link to the text. There are grammar and syntax errors that need to be improved. Thorough proof reading is suggested before resubmitting to another journal. 

Author Response

The authors are really thankful to you for providing really valuable feedback to improve the quality of the manuscript.

According to your feedback, we have made the necessary changes in the manuscript.

1. Kindly fetch the following STROBE checklist.

2. We have revised the manuscript for grammar check.

3. The point regarding only female participants: As the sample is collected from female applied medical college. Hence, Female participants were included in the study. (We have mentioned this in the manuscript)

4. You can appreciate the changes highlighted in red color in the revised manuscript.

Kind regards

Round 2

Reviewer 2 Report

Thank you for your revisions and for the chance to review this manuscript. There has been an overall positive change to your manuscript following the peer-review process, however, there are still a couple of minor details that need addressing before considering for publication.

Results: as indicated in the original, first review, please consider how best to present your results in a way that avoids long form tables such as table 3. Results from table 3 could be incorporated in text or in a more succinct table form.

Grammatical errors: even though the manuscript is greatly improved, the odd error is still present. For example, in your acknowledgements you say "we thanks...." which is incorrect. Just a quick, fresh review of the manuscript could help to get rid of these few remaining issues.

Greatly improved paper, few remaining queries before it is ready for publication. 

Author Response

The authors are thankful for the positive suggestions provided by the reviewer.

We had made the necessary changes. We have modified Table 3 into paragraphs and thus presented it in a simpler way. The changes made in revisions can be appreciated in red color. 

A fresh review of English grammar has been done. 

Regards